# Practice and factors associated with active management of third stage of labor among obstetric care providers in Amhara region referral hospitals, North Ethiopia, 2018: A cross sectional study

**Daniel Adane**[1]*, **Getahun Belay**[2ʘ], **Azimeraw Arega**[2ʘ], **Biresaw Wassihun**[3‡], **Getnet Gedefaw**[4ʘ], **Kassahun Gebayehu**[5‡]

1 Department of Midwifery, College of Medicine and Health Sciences, Wolkite University, Wolkite, Ethiopia, 2 Department of Midwifery, College of Medicine and Health Sciences, Bahir Dar University, Bahir Dar, Ethiopia, 3 Department of Midwifery, College of Medicine and Health Sciences, ArbaMinch University, Arba Minch, Ethiopia, 4 Department of Midwifery, College of Health Sciences, Woldia University, Woldia, Ethiopia, 5 Department of Nursing, College of Medicine and Health Sciences, University of Gondar, Gondar, Ethiopia

ʘ These authors contributed equally to this work.
‡ These authors also contributed equally to this work.
* danieladane178@yahoo.com

## Abstract

### Background

Active management of third stage of labor is the most indispensable intervention to avert post-partum hemorrhage which is one of the typical causes of maternal morbidity and mortality. Therefore, the aim of the study was to assess practice and factors associated with active management of third stage of labor among obstetric care providers in referral hospitals.

### Methods

Institution based cross-sectional study design was conducted from April 1–30, 2018. Simple random sampling technique was used to select a total of 356 obstetric care providers. Data were collected using pretested, structured and self-administered questionnaires. Data were entered to Epi data version 3.1 statistical software and exported to SPSS 23 for analysis. Bivariate and multivariate logistic regression analyses were performed to identify associated factors. P value <0.05 with 95% confidence level were used to declare statistical significance.

### Result

This study revealed that practice of active management of third stage of labor was 61.2%. Age group of 20–30 years [AOR = 1.95 (95%CI;1.13–3.38)], Being male obstetric care provider [AOR = 1.74 (95%CI;1.03–2.94)], having work experience ≥2 years [AOR = 1.95(95% CI;1.13–3.38)], availability of oxytocin [AOR = 5.46 (95%CI; 2.41–12.3)], having exposure to

**Data Availability Statement:** All relevant data are within the manuscript and Supporting Information files.

**Funding:** The study was funded by Bahir Dar University. The funder had no role in study design, data collection and analysis, decision to publish, or preparation of the manuscript.

**Competing interests:** The authors have declared that no competing interests exist.

**Abbreviations:** AMTSL, Active Management of Third Stage of Labor; AOR, Adjusted Odds Ratio; CCT, Control Cord Traction; COR, Crude Odds Ratio; PPH, Postpartum Hemorrhage.

manage third stage of labor [AOR = 2.91(95%CI; 1.55–5.48)], and having good knowledge [AOR = 2.67 (95%CI; 1.46–4.90)], were the factors associated with practice.

## Conclusion

This study showed that practice of active management of third stage of labor was high. Age group between 20–30 years, being a male obstetric care provider, having ≥2years work experience, availability of oxytocin, exposure to third stage management and having good knowledge were factors associated with practice. Therefore, all referral hospitals and concerned bodies need efforts to focus on providing training to increase health care provider's knowledge so as to sustain good practice through appropriate interventions.

## Introduction

Third stage of labor is the period following from the delivery of the fetus to the expulsion of the placenta and membranes. Third stage of labor is the shortest period, but critical time for maternal morbidity and mortality[1]. Active management of third stage of labor includes three interrelated but independent components such as, prophylactic administration of an uterotonic drugs (oxytocin), controlled cord traction and uterine massage [2]. Currently, World Health Organization (WHO) recommends active management of the third stage of labor as a critical intervention for postpartum hemorrhage (PPH) prevention which consequently decreases the occurrence rate of PPH by 60–70% [2].

Active management of third stage of labor (AMTSL) is defined as the stage of labor from the delivery of the fetus to the expulsion of the placenta and membranes, which includes the following AMTSL components such as: prophylactic administration of an uterotonic drugs (oxytocin), controlled cord traction and uterine massage [2].

Globally, more than half million women die as a result of pregnancy and child birth related complications[3]. Hemorrhage accounts more than 50% of direct causes of maternal deaths in the world, the death occurs typically in the postpartum period and most of them are due to PPH which occurs in low-income countries, where there are no birth attendants or where birth attendants lack the necessary skills or equipment to prevent and manage PPH and shock[4].

In Africa, due to increased prevalence risk factors such as grand-multiparty, no routine use of prophylaxis against obstetric hemorrhage combined with poorly developed obstetric services, obstetrics hemorrhage is responsible for 30% of the total maternal deaths. Sub-Saharan Africa alone accounts nearly 66% of maternal death because of poorly developed facilities and lack of trained attendants at delivery, high proportions of death occur in low income countries. Majority of these deaths occur within few hours after delivery and in most cases are due to PPH[5, 6].

Ethiopian government and Federal ministry of health (FMOH) are committed to achieve access and strengthening facility-based maternal and newborn services, and a Health Sector Development Plan (HSDP) with an aim of reducing the maternal mortality ratio (MMR) by three quarters[7]. Currently AMTSL is considered as an important component of health care provider quality assurance program [8]. For the quality of maternal care service utilization it is important to assess the existing care providers practice and associated factors. Even though few studies were conducted in Ethiopia nostudy was conducted in this region on health care providers practice and its contributing factors towards AMTSL, therefore this study aimed to assess the practice of obstetric care providers and factors associated towards AMTSL in Amhara region referral hospitals, North Ethiopia [9, 10].

## Methods

### Study area

This study was conducted in Amhara region referral hospitals. Amhara region is one of the nine regional states in Ethiopia and found in Northern part of Ethiopia. According to regional Health Bureau 2018 the region has five referral hospitals including University of Gondar Referral Hospital, Felegahiwot Referral Hospital, Dessie Referral Hospital, Debremarkos Referral Hospital and Debrebirhan Referral Hospital. They provide services over 5 million people.

### Study design and period

Institution based quantitative cross-sectional study was conducted from April 1–30, 2018

### Source population

All obstetric care providers who were working in labor ward of Amhara region referral hospitals

### Study population

All obstetric care providers who were working and managing AMTSL during the study period

### Sample size determination

A single proportion formula was used to estimate the sample size required for the study. A sample size was estimated using Epi-info version 7 software, using the prevalence of good practice on AMTSL (32.8%) [9], 95% of confidence, 5% margin of error. By considering 5% non-response, the final sample size was 356.

### Sampling procedure

The study was conducted at Amhara region referral hospitals. In this study area there are five public referral hospitals. The samples were proportionally allocated to each referral hospitals based on the number of obstetric care providers in the respective referral hospitals. We allocated sample to the respective hospitals: Felegahiwot Referral Hospital 90, Gondar Referral Hospital 127, Dessie Referral Hospital 68, Debremarkos Referral Hospital 33 and Debrebirhan Referral Hospital 38. Simple random sampling technique was used to select the study participants.

### Operational definitions

Obstetric care providers are professionals who were licensed and registered professional to give obstetric care

Active management of third stage of labor (AMTSL) is defined as the stages of labor from the delivery of the fetus to the expulsion of the placenta and membranes, which includes the following AMTSL components such as; prophylactic administration of an uterotonic drugs (oxytocin), controlled cord traction and uterine massage [2].

Health care providers who scored ≥ mean score of AMTSL practice questions is considered as having good practice. Whereas, health care providers who scored below mean score of AMTSL practice questions were considered as having poor practice.

Exposure to third stage management means obstetric care providers who have attended previously which is before the actual data collection meaning that it tell us indirectly the working experience of an individual on AMTSL.

## Data collection tools

A pretested, structured self-administered and observational checklist questionnaire was prepared based on reviewing relevant literatures. The instrument was pretested for its reliability. The content validity of the questionnaire was reviewed by qualified obstetricians and public health specialists. The questionnaire was designed in English. All obstetric care providers who were working in referral hospitals at labor wards and fulfilled eligibility criteria were included in the study. In data collection process 10 data collectors (BSc midwives) supervised by 5 (BSc Midwives having TOT on Basic emergency management and newborn care) were involved.

## Data quality control

Both interview and observation was used on the same participant. All Data collectors were working outside the study area. Before starting the actual data collection, one day training was given for both data collectors and supervisors on objectives, approach to study subjects and how to use the questionnaire. Pretest was conducted 5% of the total sample size in Debretabor general hospital among obstetric care providers to validate, assess the clarity and completeness of the tools. The reliability of the questionnaires was checked via SPSS by reliability index measurement for practice questions (Cronbach's alpha) which was 0.81. During data collection data collectors were first observe at least two deliveries while care providers practices third stage based on checklist and they would ask the same participants and supervision was done by field supervisors and over all activities was controlled by principal investigator. Finally after data collection before analysis all collected data were checked for completeness.

## Data processing and analysis

Data were coded, cleaned, edited and entered intoEPI data version 3.1 and exported to SPSS version 23.0 for statistical analysis. Descriptive statistical analysis was carried out to compute frequency, percentage and the mean for independent and dependent variables. Binary logistic regression analysis was used to ascertain the association between explanatory and outcome variables. Variables with significant (P< 0.25) association in bivariate analysis were entered into multivariate analysis and those variables with P<0.05 were considered statistically significant. The values were coded as "1 = Correct response (consistent with AMTSL components) and 0 = Incorrect response (inconsistent with AMTSL components)". Finally, a composite variable from these questions was generated to categorize obstetric care providers as having "good/poor practice. Lastly, study participants who scored mean and mean were categorized as having good practice on AMTSL. Multi-collinearity was checked to see the linear correlation among the independent variables by using standard error. Variables with a standard error of > 2 were dropped from the multivariate analysis. Model fitness was checked with Hosmer-Lemeshow test.

## Ethical approval and consent to participant

Ethical clearance was obtained from the Institutional Review Board of Bahir Dar University and then Amhara National Regional State Health Bureau wrote formal letter to all referral Hospitals of the Region and permission was taken from each Hospitals. After the purpose and objective of the study have been informed, informed verbal consent was obtained from each

study participants. All the study participants were informed about the purpose of the study. Their information was kept confidential by excluding their names in the questionnaire and by observing them alone during the observation.

## Result

### Socio-demographic characteristics of respondents

In this study, 356 obstetric care providers were participated with response rate of 100%. The mean age of the study participants were27.71 (SD± 2.95) with a range of 20–40 years. Majority of the study participants were males, 201(56.5%). Study participants with the age range of 20–30 years accounted for 313 (87.9%) (**Table 1**).

### Provider, institutional and supplies/logistics related characteristics

In this study 214(60.1%) of respondents had previous AMTSL related training after graduation to be more qualified in obstetrics practice. Majority 256(71.95) of respondents reported that they had had exposure to manage third stage of labor (**Table 2**).

### Knowledge of obstetric care providers

Two hundred fifty four (71.3%) of respondents were knowledgeable on AMTSL with the mean score of 7.14(SD = 1.49). About 290(81.5%) of respondents knew all basic components of AMTS (**Table 3**).

### Practice of obstetric care providers towards active management of third stage of labor

The result of the study revealed that, 218(61.2%) of respondents had good practice, whereas 138(38.8%) of respondents had poor practice with a mean score of practice 21.16 (SD = 2.77). Ninety eight of respondents were administering the right dose of oxytocin for AMTSL management. Three hundred fourteen (88.2%) obstetric care providers performed controlled cord traction per protocol (**Table 4**). Obstetric care providers who responded ≥ mean of the practice questions were considered as having good practice.

### Factors associated with practice of obstetric care providers towards AMTSL

Binary Logistic regression was performed to assess the association of each independent variable with practice. The result of this study revealed that age, sex, marital status, year of graduation, work experience, AMTSL related training, availability of oxytocin drugs, having exposure to manage third stage of labor and knowledge were significantly associated with practice in bivariate analysis.

In multivariate logistic regression age, sex, work experience, availability of oxytocin drugs, having exposure to manage third stage of labor and knowledge were significantly associated with practice at P-value of <0.05.

Obstetric care providers who had exposure of AMTSL had 2.67 times (AOR = 2.91; 95%CI (1.55, 5.48)) higher odds of good AMTSL practice than their counterparts.

Obstetric care providers who had good knowledge of AMTSL had 2.67 times (AOR = 2.67; 95%CI (1.46, 4.9)) higher odds of good AMTSL practice than those who had poor knowledge.

Obstetric care providers who had age range of 20–30 years had 3.86 times (AOR = 3.86; 95%CI (1.47, 10.12)) higher odds of good AMTSL practice than their counterparts.

Male obstetric care providers had about 1.74 times (AOR = 1.74; 95% CI (1.03, 2.94)) higher odds of good practice than female.

**Table 1. Socio-demographic characteristic of respondents.**

| Variables | Frequency | Percent |
|---|---|---|
| **Age** | | |
| 20–30 years | 313 | 87.9 |
| >30 years | 43 | 12.1 |
| **Gender** | | |
| Male | 201 | 56.5 |
| Female | 155 | 43.5 |
| **Marital status** | | |
| Single | 196 | 55.1 |
| Married | 160 | 44.9 |
| **Monthly income** | | |
| 1651–3145 ETB(Ethiopian birr) | 13 | 3.7 |
| 3146–5195 ETB | 189 | 53.1 |
| 5196–7758 ETB | 107 | 30.0 |
| 7,759–10,833 ETB | 39 | 11.0 |
| >10,833 ETB | 8 | 2.2 |
| **Ethnicity** | | |
| Amhara | 240 | 67.4 |
| Oromia | 50 | 14.1 |
| Tigrie | 26 | 7.3 |
| SNNPR | 31 | 8.7 |
| Other* | 9 | 2.5 |
| **Qualification** | | |
| Diploma | 42 | 11.8 |
| BSc | 259 | 72.8 |
| MSc | 10 | 2.8 |
| Resident | 37 | 10.4 |
| Obstetrician | 8 | 2.2 |
| **Religion** | | |
| Orthodox Christian | 303 | 85.1 |
| Muslim | 32 | 9.0 |
| Protestant | 20 | 5.9 |
| **Profession** | | |
| Medical intern | 100 | 28.1 |
| Midwife | 211 | 59.3 |
| Obstetric resident | 37 | 10.4 |
| Obstetrician | 8 | 2.2 |
| **Year of graduation** | | |
| 2016–2018 | 220 | 85.94 |
| <2016 | 36 | 14.06 |
| **Work experience** | | |
| 0–2 years | 156 | 43.8 |
| ≥ 2 years | 200 | 56.2 |

*other = = (Afar, Somalia)

Obstetric care providers who had more than 2 years' work experience had 1.95 times (AOR = 1.95; 95% CI (1.13, 3.38)) higher odds of good practice than female.

**Table 2. Provider, institutional and supplies related characteristic of respondents.**

| Variables | Frequency | Percent |
|---|---|---|
| **AMTSL related training** | | |
| Yes | 214 | 60.1 |
| No | 142 | 39.9 |
| **Availability of oxytocin drugs** | | |
| Yes | 305 | 85.7 |
| No | 51 | 14.3 |
| **AMTSL is important** | | |
| Yes | 339 | 95.2 |
| No | 17 | 4.8 |
| **Personally conduct AMTSL** | | |
| Yes | 335 | 94.1 |
| No | 21 | 5.9 |
| **Frequent practice of third stage of labor** | | |
| Yes | 256 | 71.9 |
| No | 79 | 22.1 |

**Table 3. Knowledge of obstetric care providers.**

| Variables | Frequency | Percent |
|---|---|---|
| **Knowing of uterotonic drugs** | | |
| Oxytocin | 77 | 21.6 |
| Ergometrine | 24 | 6.7 |
| Misoprostol | 7 | 2.0 |
| All | 248 | 69.7 |
| **Knowing dose of oxytocin** | | |
| 10 IU | 323 | 90.7 |
| 5 IU | 33 | 9.3 |
| **Knowing route of oxytocin** | | |
| IM | 326 | 91.6 |
| IV | 30 | 8.4 |
| **Role immediately after delivery of baby** | | |
| Check presence of other baby | 256 | 71.9 |
| Administer uterotonic drugs | 85 | 23.9 |
| Uterine massage | 12 | 3.4 |
| Apply controlled cord traction | 3 | 0.8 |
| **Time of administration of uterotonic drug** | | |
| Within 1minute after delivery of the baby | 306 | 86.0 |
| After delivery of anterior shoulder of the baby | 35 | 9.8 |
| Within 2–3 minute after delivery of the baby | 9 | 2.5 |
| >3 minute after delivery of the baby | 6 | 1.7 |
| **Mention essential components of AMTSL** | | |
| Administer Uterotonic drugs | 45 | 12.6 |
| Apply controlled cord traction | 7 | 2.0 |
| Uterine massage | 14 | 3.9 |
| All | 290 | 81.5 |
| **Knowledge of care providers** | | |
| Good knowledge | 254 | 71.3 |
| Poor knowledge | 102 | 28.7 |

Table 4. Practices of the obstetric care providers.

| Variables | Frequency | Percent |
|---|---|---|
| **Palpates mothers abdomen immediately after delivery of the first baby** | | |
| Yes | 250 | 70.2 |
| No | 106 | 29.8 |
| **Uterotonic drugs given** | | |
| Oxytocin | 339 | 95.2 |
| Ergometrine | 17 | 4.8 |
| **Dose of oxytocin given** | | |
| 10 IU | 349 | 98.0 |
| 5 IU | 7 | 2.0 |
| **Route of oxytocin given** | | |
| IM | 345 | 96.9 |
| IV | 11 | 3.1 |
| **Time of oxytocin given** | | |
| Within 1$^{st}$ min | 314 | 88.2 |
| Within 2–3 min | 42 | 11.8 |
| **Wait uterine contraction 2–3 min to apply CCT** | | |
| Yes | 185 | 52.0 |
| No | 171 | 48.0 |
| **CCT applied** | | |
| Yes | 314 | 88.2 |
| No | 42 | 11.8 |
| **Placenta supported with both hands during placenta delivery** | | |
| Yes | 303 | 85.1 |
| No | 53 | 14.9 |
| **Membrane extracted gently with lateral movement** | | |
| Yes | 286 | 80.3 |
| No | 70 | 19.7 |
| **Uterine massage immediately after delivery of placenta** | | |
| Yes | 300 | 84.3 |
| No | 56 | 15.7 |
| **Uterine relaxation was ensured** | | |
| Yes | 252 | 70.8 |
| No | 104 | 29.2 |
| **Inform and demonstrate the mother how to massage uterus** | | |
| Yes | 239 | 67.1 |
| No | 117 | 32.9 |
| **Practice** | | |
| Good practice | 218 | 61.2 |
| Poor practice | 138 | 38.8 |

Obstetric care providers who had oxytocin drugs in the ward had 5.46 times (AOR = 5.46; 95% CI (2.41, 12.3)) higher odds of good practice than who had no oxytocin drugs (**Table 5**).

## Discussion

This study showed that 61.2% of respondents had good practice with 95% CI (55.9–66.6). The finding of this study is higher than the study conducted in Egypt, Tanzania, South Nigeria,

**Table 5. Factors associated with practice among obstetric care providers.**

| Variables | Practice | | OR(95% CI) | |
|---|---|---|---|---|
| | Good | Poor | COR | AOR |
| **Age** | | | | |
| 20–30 | 128 | 185 | 2.28(1.09–4.780) | 3.86(1.47–10.12)** |
| >30 | 10 | 33 | 1 | 1 |
| **Gender** | | | | |
| Male | 137 | 64 | 1.96 (1.27–3.02) | 1.74(1.03–2.94)* |
| Female | 81 | 74 | 1 | 1 |
| **Marital status** | | | | |
| Married | 107 | 53 | 1.55 (1.00–2.39) | 1.18(0.65–2.11) |
| Single | 111 | 85 | 1 | 1 |
| **Year of graduation** | | | | |
| Before 2016 | 20 | 16 | 0.58 (0.29–1.19) | 1.1 (0.44–2.63) |
| 2016–2018 | 150 | 70 | 1 | 1 |
| **Work experience** | | | | |
| ≥2 years | 139 | 61 | 2.22(1.44–3.43) | 1.95(1.13–3.38) * |
| 2 years | 79 | 77 | 1 | 1 |
| **Taking AMTSL related training** | | | | |
| Yes | 142 | 72 | 1.71 (1.11–2.65) | 0.96(0.53–1.74) |
| No | 76 | 66 | 1 | 1 |
| **Availability of oxytocin** | | | | |
| Yes | 203 | 102 | 4.78 (2.50–9.13) | 5.46(2.41–12.3)*** |
| No | 15 | 36 | 1 | 1 |
| **Having exposure of AMTSL** | | | | |
| Yes | 181 | 75 | 4.40(2.58–7.50) | 2.91(1.55–5.48)** |
| No | 28 | 51 | 1 | 1 |
| **Knowledge** | | | | |
| Good | 181 | 73 | 4.36(2.68–7.09) | 2.67(1.46–4.90)** |
| Poor | 37 | 65 | 1 | 1 |

*p = = <0.05

** = = p <0.01

*** = = = p<0.001

SidamaZone, Hawassa City and Addis Ababa, with the prevalence of practice 15%, 7%, 41%, 32.8%, 16.7% and 47% respectively [9, 10, 11,12,13, 14]. This difference might be due to the difference in study area. This study was conducted at tertiary health institutions, which is probably the obstetric care providers on these facilities might have practiced AMTSL under supervision of respective senior care providers than primary healthcare facilities.

Obstetric care providers whose age groups of 20–30 years were 3.86 times more likely practice AMTSL than others. This might be due to that obstetric care providers in 20–30 age ranges may have good knowledge because of the year of graduation is not too far, their skill and practice recall bias is minimal as a result they can practicing AMTSL easily as compared to their counter parts.

The sex of the obstetric care providers was found to be positively associated with practice on AMTSL. Male obstetric care providers were 1.74 times more likely practiced AMTSL than females. This might be due to more than half of the respondents and those who had good practice towards AMTSL were male obstetric care providers.

Work experience ≥2 years were 1.95 times more likely to practice AMTSL than others who had work experience <2 years. The finding of this study was consistent with the study done in Sidama Zone, and Addis Ababa, Ethiopia [9–10]. This may be the more they stay in professional work they might have training, more experience as a result obstetric care providers competency on AMTSL is increasing.

Health care providers who had oxytocin drugs in the ward were 5.46 times more likely practiced AMTSL than their counter parts. The finding of this study was supported by the study done in Nigeria [14]. This might be due to an increase in case flow which may increase the need of oxytocin drugs. Even though the availability of the drug depends on the supply of the hospitals, marked shortage is encountered in low income countries which may inhibit proper practice.

Obstetric care providers who had having exposure to manage third stage of labor were 2.91 times more likely practiced AMTSL than who had no exposure to manage AMTSL. This is explained as obstetric care providers who were practicing frequently, can manage AMTSL by implementing the basic components of the AMTSL. Moreover, Professionals who had advanced training besides their licensed qualification, has tremendous effect on AMTSL management.

Obstetric care providers who had good knowledge were 2.67 times more likely practiced AMTSL than its counter parts, which was consistent with the study conducted in other parts of Ethiopia [9, 10]. This might be due to that if the obstetric care providers have poor knowledge, they are less likely to implement the standard practice of AMTSL which may affect the health condition of the mother and the baby.

## Conclusion

This study showed that practice of active management of third stage of labor was high. Age, gender, work experience, availability of oxytocin drugs, exposure to third stage management and having good knowledge were some of significant factors associated with practice. Therefore, all referral hospitals and other concerned bodies need efforts to focus on to increase infrastructure and providing training to increase health care provider's knowledge so as to sustain good practice through appropriate interventions.

## Supporting information

**S1 File. This is the S1 file SPSS data set.**
(SAV)

## Acknowledgments

The authors would like to thank Bahir Dar University and Amhara public health institute for permitting ethical clearance and giving permission letter to do this research respectively. Our gratitude goes to data collectors, respondents who participated in this study.

## Author Contributions

**Conceptualization:** Daniel Adane, Getahun Belay, Azimeraw Arega, Biresaw Wassihun, Getnet Gedefaw, Kassahun Gebayehu.

**Data curation:** Daniel Adane, Biresaw Wassihun, Getnet Gedefaw.

**Formal analysis:** Daniel Adane.

**Methodology:** Daniel Adane, Getahun Belay, Azimeraw Arega, Biresaw Wassihun, Getnet Gedefaw, Kassahun Gebayehu.

**Supervision:** Daniel Adane.

**Writing – original draft:** Daniel Adane.

**Writing – review & editing:** Daniel Adane, Getahun Belay, Azimeraw Arega, Biresaw Wassihun, Kassahun Gebayehu.

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
