## [Decision Letter · Decision Letter 0]

14 Aug 2019

PONE-D-19-18594

Practice and factors associated with active management of third stage of labor among obstetric care providers in Amhara region referral hospitals, North Ethiopia, 2018: A cross sectional study.

PLOS ONE

Dear Mr Adane,

Thank you for submitting your manuscript to PLOS ONE. After careful consideration, we feel that it has merit but does not fully meet PLOS ONE’s publication criteria as it currently stands. Therefore, we invite you to submit a revised version of the manuscript that addresses the points raised during the review process.

We would appreciate receiving your revised manuscript by Sep 28 2019 11:59PM. To enhance the reproducibility of your results, we recommend that if applicable you deposit your laboratory protocols in protocols.io, where a protocol can be assigned its own identifier (DOI) such that it can be cited independently in the future. For instructions see: http://journals.plos.org/plosone/s/submission-guidelines#loc-laboratory-protocols

We look forward to receiving your revised manuscript.

Kind regards,

Charles A. Ameh, PhD, MPH, FWACS (OBGYN), FRCOG

Academic Editor

PLOS ONE

3. Please include additional information regarding the survey or questionnaire used in the study and ensure that you have provided sufficient details that others could replicate the analyses. For instance, if you developed a questionnaire as part of this study and it is not under a copyright more restrictive than CC-BY, please include a copy, in both the original language and English, as Supporting Information.  If the original language is written in non-Latin characters, for example Amharic, Chinese, or Korean, please use a file format that ensures these characters are visible."" "

4. Please state whether you validated the questionnaire prior to testing on study participants. Please provide details regarding the validation group within the methods section

5. Please amend your current ethics statement to address the following concerns: Please explain why was written consent was not obtained, how you recorded/documented participant consent, and if the ethics committees/IRBs approved this consent procedure.

6. Thank you for stating the following in the Acknowledgments Section of your manuscript: "The authors thank Bahir Dar University for approval of ethical clearance, technical and financial

support of this study"

Additional Editor Comments (if provided):

Thanks for submitting an interesting article for consideration. The manuscript requires a major revision and a complete proofreading/editing by a native English language writer before it can be considered for publication. Please see specific comments by both reviewers.

Reviewers' comments:

Reviewer's Responses to Questions

**Comments to the Author**

1. Is the manuscript technically sound, and do the data support the conclusions?

Reviewer #1: Yes

Reviewer #2: Partly

2. Has the statistical analysis been performed appropriately and rigorously? 

Reviewer #1: Yes

Reviewer #2: Yes

3. Have the authors made all data underlying the findings in their manuscript fully available?

Reviewer #1: Yes

Reviewer #2: Yes

4. Is the manuscript presented in an intelligible fashion and written in standard English?

Reviewer #1: Yes

Reviewer #2: No

5. Review Comments to the Author

Reviewer #1: The study is commendable however there are some issues that needs to be addressed before publication. This is contained in my comments which I have sent as an attachment.

Reviewer #2: The paper has so much grammatical issues that make understanding the concept difficult.

I also noted that the aim of the study is clear in the abstract, but not quite the same in the body of the work, and less comprehensible.

The second paragraph in the background section defines active management of the third stage of labour; this definition looks incomplete.

The same applies to the second paragraph on the Operational definitions section when defining active management of the third stage of labour.

On page 8, the first paragraph of “Provider, institutional and supplies…….” This paragraph was not flowing and not comprehensible. Some sentences were incomplete.

Page 13, second paragraph, the discussion on age and gender of the participant is difficult to understand

6. PLOS authors have the option to publish the peer review history of their article (what does this mean?). If published, this will include your full peer review and any attached files.

Reviewer #1: Yes: DR HAJARATU UMAR SULAYMAN

Reviewer #2: Yes: Rakiya Saidu

---

## [Author Response · Author response to Decision Letter 0]

20 Aug 2019

Author’s Point-by-Point Response to the Reviewer's and Editors 

Title: Practice and factors associated with active management of third stage of labor among obstetric care providers in Amhara region referral hospitals, North Ethiopia, 2018

Corresponding author: Daniel Adane/danieladane178@yahoo.com

Authors

1. Daniel Adane/danieladane178@yahoo.com

2. Getahun Belay/geichbelay@gmail.com

3. Azimeraw Arega/wazme84@gmail.com

4. Biresaw Wassihun/bireswas@gmail.com

5. Getnet Gedefaw/gedefawget@gmail.com

6. Kassahun Gebayehu/kassish6@gmail.com

Manuscript #: PONE-D-19-18594

Journal: PLOS ONE 

Article type: Research article

Point by point response to Reviewers and Editor

First of all, the authors would like to thank Plos one Journal editors and the respective reviewers for reviewing our manuscript and providing the necessary comments to be corrected. As per the comments given, we have made corrections point by point to comment. The authors tried to answer all the issues raised by editorial team and reviewers. 

Point by point response to Editor

Dear Dr. Charles A. 

We wrote the manuscript based on the PLOS ONE journal submission guideline

1. Please include additional information regarding the survey or questionnaire used in the study and ensure that you have provided sufficient details that others could replicate the analyses. For instance, if you developed a questionnaire as part of this study and it is not under a copyright more restrictive than CC-BY, please include a copy, in both the original language and English, as Supporting Information. If the original language is written in non-Latin characters, for example Amharic, Chinese, or Korean, please use a file format that ensures these characters are visible."" 

Response: Thank you very much. The questionnaires were in English version and we can attach as supplementary files.

2. Please state whether you validated the questionnaire prior to testing on study participants. Please provide details regarding the validation group within the methods section

Response: Thank you very much. We explained in the method section as per your constructive suggestion and comments 

3. Please amend your current ethics statement to address the following concerns: Please explain why was written consent was not obtained, how you recorded/documented participant consent, and if the ethics committees/IRBs approved this consent procedure.

Response: Since the study is on the professional who has no patient outcome, IRB was deciding to do with informed consent. We recorded the document of the study participants by the data collector since the questioners has choice before data collection whether they are volunteer to participate or not at the time of data collection then the data collectors inform the study participants about the objective, the importance and everything about the research. The study participants were responded by agree or disagree then the data collector will tick I agree box if the participants are volunteer to participate in the study and I don’t agree box if participants are not volunteer to participate in the questionnaires (you can see the questionnaires which is attached as supplementary file). 

4. Thank you for stating the following in the Acknowledgments Section of your manuscript:" The authors thank Bahir Dar University for approval of ethical clearance, technical and financial

support of this study"

Response: Thanks in depth on your invaluable and concrete comments from the beginning to now. We amended it and we rewrote it again in main text.

Point by point response to Reviewer # 1

Dear, DR HAJARATU UMAR SULAYMAN 

Question 1: The overall aim of this study was said to be “to fill the gaps and provide good intervention for care providers on practice and associated factor towards AMTSL at institution level”. This was however not written clearly under a separate heading. This study by the nature of it’s design may not be able to provide good intervention, the author may need to look at the aim again.

Response 1: Thank you very much. We amended it in the main text. Of course you are right this study aim was not to provide good intervention rather we assessed the practice and associated factors of AMTSL among obstetric care providers. 

Question 2: Under the Results, the labelling of the tables had the title of the whole study repeated under each table. My suggestion would be that Table 1 should be titled: Sociodemographic characteristics of the respondents, Table 2 should be: Provider, Institutional and Supplies related characteristics of respondents, and so on.

Response 2: Thanks alot .We incorporated and we rewrote it again based on your suggestion in the main text.

Question3: In Table 1, Under the Variables: in the subtitle of Qualifications; Senior Gynaecologist was used, I suggest that this be changed to Senior Obstetrician as this was an Obstetric study. Same goes for the subtitle of Profession where the word Gynaecologist was used instead of Obstetrician. Under the subtitle of Religion does Orthodox mean Catholic? Since the Protestants and Muslims both fall under orthodox religions. Medical Intern appeared under both Year of Graduation as well as under Profession. Please clarify.

Response 3: Thank you very much. We clarified in the paper. We changed the professional name from Gynecologist to obstetrician. Regarding religion we have tried to clarify, In Ethiopia there are different people who follow different type of Christian religion. Christian religion categorized in to different religion such as orthodox Christian, protestant Christian, catholic Christian, and Adventist Christian. Therefore in our study there were people who follow Muslim religion, orthodox Christian and protestant. We haven’t gotten another religion follower during our study period. We apologize medical interns are out of classification of year of graduation which is typing error since they are practicing, they are not graduated professionals but they grouped under profession.

Question 4: Immediately after Table 1, under the sub heading “Provider, institutional and supplies/logistics related characteristics” in the first sentence, it was written that: “In this study 214(60.1%) of respondents were taken AMTSL related training”. This sentence is not clear. Do you mean the respondents had a previous training in AMTSL or are presently undergoing such training? Kindly clarify. 

Response 4: Thank you in depth. In this study 214(60.1%) of respondents were taken AMTSL related training meaning that the respondents had a previous training in AMTSL, so we have seen it again and we correct it as ‘’In this study 214(60.1%) of respondents had a previous AMTSL related training’’. The questions focus on in addition to their professional competency, we assessed whether they have taken additional AMTSL related training or not. Therefore it doesn’t mean that they are presently undergoing such training

Question 5: In Table 2: Under Variables: Does accessibility mean that these drugs were available for use at the time when it was needed? Accessibility is not synonymous to availability. Please explain.

Response 5: Thank you. We amended it in the main text simply we asked them about the presence of the drug which is availability. Therefore we changed the word to availability

Question 6: In Table 3: “Types of Variables” was used as against” Variables” used in Tables 1 & 2, please ensure consistency. Under the variable “Knowing the route of Oxytocin,”is it possible to know both the im and the iv routes? That option was not given.

Response 6: Thank you alot. We changed types of variables to “variables” and for the route of administration, we had the option “both” during the data collection, but we haven’t got any response for both (IM and IV route). That is why we are writing the two categorical variables under knowing the route of oxytocin for AMTSL since the frequency for both route was zero.

Question 7: Under Time of administration of uterotonic drugs; do you mean 1 minute/ 2-3 minutes of delivery of the baby? Clarify. Same in Table 4. 

Response 7: Thank you very much. Actually different obstetric books recommended giving oxytocin after the anterior shoulder of the fetus is delivered. According to FIGO/ICM recommendations the recommended time for administration of uterotonic drugs is immediately within 1 minute after delivery of the baby similar with our national obstetrics protocol encourages giving uterotonic drugs within 1minute of delivery. We searched different protocols and literatures to classify the following four categories under the time of administration of uterotonic drug: Within 1 minute, after delivery of anterior shoulder, within 2- 3 minute and 3 minute after delivery of placenta because a second of time has business on obstetric hemorrhage.

 Question 8: In Table 4 how was uterine relaxation ensured?

Response 8: Thank you in advance. We ensured whether the uterus is relaxed or well contracted via abdominal palpation immediately giving birth. Uterine relaxation can be ensured by palpating mother’s abdomen so as to know whether the uterus is contracted or relaxed (if the uterus is hard round on palpation it is shows contracted uterus, otherwise it is relaxed), because once the uterus is relaxed the mother will be at risk of PPH.

 If we got hard round at umbilicus which is around 20 weeks of gestation without excessive vaginal bleeding which causes vital sign derangement, we declared the uterus is well contracted and we told the mother to massage her uterus every 15 minutes for the following 2 hours. If the above scenario is not occurred the uterus is relaxed and we are going to manage for uterine atony. Therefore the terminology of uterine relaxation is to indicate uterine atony.

Question 9: In the beginning of the second sentence in page 12 the following was written: “In multivariable logistic regression…..” Do you mean multivariate logistic regression? Kindly explain.

Response 9: Thank you. We amended it in the main text.

 Question 10: In Page 12 it is not clear where the discussion started from since there is a table immediately below the heading “Discussion”. The table should be under the results.

Response 10: Thanks in advance!. We made the discussion clear. It was typing and editing error

Question 11: Table 5 is not clear. There should be a key to explain the single, double or triple asterix.

Response 11: We put the asterix and we clarified.

Question 12: In page 13 in the second paragraph the author writes that: “Similarly this study showed that male obstetric care providers were 1.74 times more likely practiced AMTSL than females, which is different from other study. This might be due to assumption that females who have even the same opportunity on interaction with laboring mothers, males can have better interaction and they will act accordingly.” This sentence is not clear. Are women not more likely to be Nurses and Midwives hence having more opportunity to practice AMTSL? This particular sentence and few other sentences have some grammatic errors. I suggest the author works with a writing coach to improve this.

Response 12: Interesting suggestion. We clarified it in revised version of manuscript.

 Point by point response to Reviewer# 2

 Dear, Rakiya Saidu 

Question 1: The paper has so much grammatical issues that make understanding the concept difficult.

I also noted that the aim of the study is clear in the abstract, but not quite the same in the body of the work, and less comprehensible.

Response 1: We would like to say thank you very much for your invaluable comments and suggestions. We considered and modified and rewrote again based on your constructive issues regarding language, coherence and comprehensibility of the manuscript 

Question 2: The second paragraph in the background section defines active management of the third stage of labour; this definition looks incomplete

Response 2: AMTSL consists only the following consecutive procedures. Checking whether additional fetus or not, administering uterotonic agent, CCT and uterine massage. We wrote by incorporating. Our references are FIGO/ICM, ACOG, RCOG, WHO.

Question 3: The same applies to the second paragraph on the Operational definitions section when defining active management of the third stage of labour.

Response 3: Same as the above and we rewrote it.

Question 4: On page 8, the first paragraph of “Provider, institutional and supplies…….” This paragraph was not flowing and not comprehensible. Some sentences were incomplete.

Response 4: Thank you very much. We re-phrased it in the manuscript.

Question 5: Page 13, second paragraph, the discussion on age and gender of the participant is difficult to understand

Response 5: Thank you for your suggestion and asking us for clarity. We did it in the main text of the manuscript.

---

## [Decision Letter · Decision Letter 1]

10 Sep 2019

[EXSCINDED]

Practice and factors associated with active management of third stage of labor among obstetric care providers in Amhara region referral hospitals, North Ethiopia, 2018: A cross sectional study.

PONE-D-19-18594R1

Dear Dr. Adane,

We are pleased to inform you that your manuscript has been judged scientifically suitable for publication and will be formally accepted for publication once it complies with all outstanding technical requirements.

With kind regards,

Charles A. Ameh, PhD, MPH, FWACS (OBGYN), FRCOG

Academic Editor

PLOS ONE

Additional Editor Comments (optional):

Thanks for addressing all the comments from both reviewers, I am pleased to accept your manuscript for publication.

Reviewers' comments:

Reviewer's Responses to Questions

**Comments to the Author**

1. If the authors have adequately addressed your comments raised in a previous round of review and you feel that this manuscript is now acceptable for publication, you may indicate that here to bypass the “Comments to the Author” section, enter your conflict of interest statement in the “Confidential to Editor” section, and submit your "Accept" recommendation.

Reviewer #1: All comments have been addressed

Reviewer #2: All comments have been addressed

2. Is the manuscript technically sound, and do the data support the conclusions?

Reviewer #1: Yes

Reviewer #2: Yes

3. Has the statistical analysis been performed appropriately and rigorously? 

Reviewer #1: Yes

Reviewer #2: Yes

4. Have the authors made all data underlying the findings in their manuscript fully available?

Reviewer #1: Yes

Reviewer #2: Yes

5. Is the manuscript presented in an intelligible fashion and written in standard English?

Reviewer #1: Yes

Reviewer #2: Yes

6. Review Comments to the Author

Reviewer #1: The Authors have addressed the concerns raised satisfactorily. I have gone through the revised paper and hereby recommend that it be published by you if it meets the specifications of your journal.

Reviewer #2: A few more grammatical issues: a few suggestions

Discussion:

Paragraph 2: "This might be because..........." may replace "This might be due to that obstetric care providers"

Paragraph 3: "the gender of the obstetric care worker........." may replace "The sex of the obstetric care"

Paragraph 4: May start with "Obstetrics care providers with more than 2 years..........."

Paragraph 7: "This might be because..........." may replace "This might be due to that.........."

7. PLOS authors have the option to publish the peer review history of their article (what does this mean?). If published, this will include your full peer review and any attached files.

Reviewer #1: Yes: DR HAJARATU UMAR SULAYMAN

Reviewer #2: Yes: Rakiya Saidu

---

## [Editor Report · Acceptance letter]

25 Sep 2019

PONE-D-19-18594R1 

Practice and factors associated with active management of third stage of labor among obstetric care providers in Amhara region referral hospitals, North Ethiopia, 2018: A cross sectional study. 

Dear Dr. Adane:

I am pleased to inform you that your manuscript has been deemed suitable for publication in PLOS ONE. Congratulations! Your manuscript is now with our production department. 

With kind regards,

on behalf of

Dr. Charles A. Ameh 

Academic Editor

PLOS ONE